# Transport signature of the magnetic Berezinskii-Kosterlitz-Thouless transition

Se Kwon Kim[1,2], Suk Bum Chung[3,4,5,6*],

**1** Department of Physics, KAIST, Daejeon 34141, Republic of Korea
**2** Department of Physics and Astronomy, University of Missouri, Columbia, MO 65211, USA
**3** Department of Physics, University of Seoul, Seoul 02504, Republic of Korea
**4** Natural Science Research Institute, University of Seoul, Seoul 02504, Republic of Korea
**5** School of Physics, Korea Institute for Advanced Study, Seoul 02455, Republic of Korea
**6** Center for Correlated Electron Systems, Institute for Basic Science (IBS), Seoul National University, Seoul 08826, Republic of Korea
* sbchung0@uos.ac.kr

September 29, 2020

## Abstract

**Motivated by recent experimental progress in 2D magnetism, we theoretically study spin transport in 2D easy-plane magnets at finite temperatures across the Berezinskii-Kosterlitz-Thouless (BKT) phase transition, by developing a duality mapping to the 2+1D electromagnetism with the full account of spin's finite lifetime. In particular, we find that the non-conservation of spin gives rise to a distinct signature across the BKT transition, with the spin current decaying with distance power-law (exponentially) below (above) the transition; this is detectable in the proposed experiment with $NiPS_3$ and $CrCl_3$.**

# 1 Introduction

Progress in the experimental detection of the celebrated Brezinskii-Kosterlitz-Thouless (BKT) phase transition has varied between the different types of physical systems. This phase transition was one of the first example of the continuous phase transition outside the Landau paradigm, involving not the symmetry breaking but rather the topological defect pair unbinding. It was theoretically formulated for the 2D XY systems [1, 2], examples of which include the 2D easy-plane magnets and the thin films of superfluids / superconductors. Experimental efforts have been devoted almost entirely to the latter, *e.g.* Refs. [3–5]. By contrast, there has been much less experimental study of the magnetic BKT transitions, not the least due to the absence of good material candidate until the recent fabrication of the monolayer van der Waals (vdW) material such as $NiPS_3$ [6–8] and $CrCl_3$ [9–11]; the latter in particular has been experimentally shown to combine a very strong easy-plane anisotropy with a nearly perfect in-plane isotropy [10]. However, experimental methods used so far to probe 2D easy-axis magnetism such as the Kerr rotation [12, 13] and the Raman spectroscopy [14, 15] detects the long-range order parameter, making them unsuitable for probing the BKT transition. Hence, to unambiguously detect the magnetic BKT transition in these materials, theoretical study of its phenomenology is required. For instance, while the transport measurements have been often used to confirm the BKT transition in 2D superconductors, *e.g.* Refs. [4, 16, 17], the transport signature of the magnetic BKT transition should be different as spin, unlike charge, is not conserved. However, this interplay of spin dissipation and the magnetic BKT transition has not been studied yet.

Related to the transport signature of the magnetic BKT transition is the issue of the long-range spin transport in 2D magnetic insulators. The spin transport via collective magnetic excitations may not show the exponential suppression in the long-distance limit that characterize the single-electron spin transport in metals. One simple example of this spin transport arises when there is a planar spiraling of the order parameter in magnetic insulators with the easy-plane anisotropy [18–21]. Given that this order parameter requires a spontaneously broken U(1) symmetry, a close analogy (summarized in Appendix A) can be developed with the superfluid transport, which can be described by the gradient of the U(1) phase of the condensate wavefunction [22, 23]. While the realization of such superfluid spin transport has been reported recently [24, 25], there remains the question whether the long-range spin ordering is a necessary condition. Given that the magnetic BKT transition, unlike the three-dimensional magnet analyzed in Ref. [26], does not arise from the long-range spin ordering, the answer to this question would determine the extent of both the impact that the magnetic BKT can have on spin transport, and the applicability of the 2D magnetic atomic monolayer to spintronics [27].

In this work, we examine the possibility of the long-distance spin transport in proximity to the magnetic BKT transition using the duality mapping from the 2D easy-plane magnetism to the electromagnetism (EM) in the $d = 2 + 1$ spacetime [28–31]. This allows us to both pursue close analogy to the current transport and pinpoint the difference that arises when the phenomenological finite spin lifetime is inserted. We find that the superfluid spin transport, *i.e.* decaying algebraically with distance, persists below the BKT temperature, while above the BKT temperature it decays exponentially with distance. In particular, we identify the vortex-induced temperature dependence of the decaying behavior of non-local spin-transport signal, which includes the previously known result for zero temperature [19] as a special case.

For the remainder, we will first review the dual $d = 2 + 1$ EM formalism and its application to the current transport near the BKT transition in the thin superconducting film; then we will discuss how this transport result is modified for spin transport in 2D XY magnets near the magnetic BKT transition due to the finite spin lifetime, together with the result for a realistic experimental setup.

## 2  Dual EM formulation of superconducting films

We first review the qualitative derivation of the transport near the BKT transition in the superconducting films using the dual $d = 2 + 1$ EM theory [28–31]. We start with the Lagrangian density,

$$\mathcal{L} = 2\pi(-na_0 + \mathbf{j} \cdot \mathbf{a}) + \frac{1}{2K}(\mathbf{e}^2 - v^2 b^2);  \tag{1}$$

$n$ and $\mathbf{j}$ are the density and the current density, respectively, of superconducting vortices, and $v$ is the dual EM wave velocity. For $\mathcal{L}$ to be useful, the dual electric ($\mathbf{e} = -\boldsymbol{\nabla}a_0 - \partial_t\mathbf{a}$) and magnetic ($b = \hat{\mathbf{z}} \cdot \boldsymbol{\nabla} \times \mathbf{a}$) fields together with the parameter $K$ need to be defined. The first step is to note the relation

$$n = \frac{\hbar}{2\pi qK}\hat{\mathbf{z}} \cdot (\boldsymbol{\nabla} \times \mathbf{J})  \tag{2}$$

(where $q = 2e$ is the charge of a single Cooper pair) between the Cooper charge current $\mathbf{J}$ and the vorticity that holds at the long-wavelength limit ($K$ is the phase stiffness). Given that we want to map vortices to particles in this formulation, a natural course is to figure out a way to make Eq. (2) equivalent to the Gauss' law. This can be accomplished by setting $\mathbf{e} = \frac{\hbar}{q}\mathbf{J} \times \hat{\mathbf{z}}$, i.e. taking the dual gauge field to originate from the Cooper pair density and current density. This concisely expresses the equivalence between the dual EM wave and the phase mode, e.g. the logarithmic vortex-vortex interaction that Eq. (1) readily yields is identical to the integration of the Cooper pair current density energy $\hbar^2\mathbf{J}^2/2q^2K$ between two vortices. The combination of the vorticity conservation $\partial_t n + \boldsymbol{\nabla} \cdot \mathbf{j} = 0$ and the divergence of the London penetration in the thin film limit, which leaves the phase mode gapless, we obtain for the vortex current

$$\mathbf{j} = \frac{\hbar}{2\pi qK}\hat{\mathbf{z}} \times \left(\frac{\partial\mathbf{J}}{\partial t} + v^2\boldsymbol{\nabla}\rho\right),  \tag{3}$$

which, by taking $b = \frac{\hbar}{q}\rho$, is the equivalent of the Ampère-Maxwell law with $\rho$ being the charge density. [1] Lastly, the Faraday's law gives us the Cooper pair current conservation,

$$0 = \boldsymbol{\nabla} \times \mathbf{e} + \frac{\partial b}{\partial t} = \frac{\hbar}{q}\left(\boldsymbol{\nabla} \cdot \mathbf{J} + \frac{\partial\rho}{\partial t}\right).  \tag{4}$$

Within the context of the dual $d = 2 + 1$ EM theory of Eq. (1), the effect of the vortex-antivortex unbinding on the superconducting film transport is most clearly manifest through the constitutive relation between $\mathbf{j}$ and $\mathbf{e}$ (that is, $\mathbf{J}$). A single vortex is phenomenologically known to have a finite mobility, i.e. $\mathbf{v} = w\mu\mathbf{e}$ where $w$, $\mu$, $\mathbf{v}$ are the winding number,

---

[1] From the fluid mechanic, the combination of the Euler equation and the gyrotropic effect $\partial\mathbf{J}/\partial t = -q\boldsymbol{\nabla}P + 2\pi qK\hat{\mathbf{z}} \times \mathbf{j}$, where $P$ is the pressure, also gives us this result by noting $v^2 = q\partial P/\partial\rho$.

the mobility and the velocity, respectively, of the vortex [32, 33]. Hence, above the BKT temperature, where a finite density of free vortices is present, the constitutive relation is

$$\mathbf{j} = \sigma_{\mathrm{dual}}\mathbf{e} = \frac{\hbar\mu}{q}n_f\mathbf{J} \times \hat{\mathbf{z}} \quad \text{for } T > T_{\mathrm{BKT}}, \tag{5}$$

where $\sigma_{\mathrm{dual}} = \mu n_f$ is the dual (or vortex) conductivity above $T_{\mathrm{BKT}}$, with $n_f$ being the combined density of free vortices and free antivortices, that vanishes singularly on approaching $T_{\mathrm{BKT}}$ as $\ln n_f \propto -1/\sqrt{T/T_{\mathrm{BKT}} - 1}$ [32, 33]. By contrast, for $T < T_{\mathrm{BKT}}$, there is no free vortex in absence of $\mathbf{J}$, so $\mathbf{j}$ arises only through the vortex-antivortex unbinding driven by $\mathbf{J}$. In this case, one part of the vortex energy arises from the Cooper pair current exerting the dual electric force, *i.e.* the $\mathbf{J} \times \hat{\mathbf{z}}$ Magnus force, on each vortex, which pushes vortices and antivortices in the opposite directions with the strength proportional to the Cooper pair current magnitude $J = |\mathbf{J}|$. The other part is the attractive vortex-antivortex logarithmic interaction, which is independent of $\mathbf{J}$. Equating these two energies give us the free energy barrier against the vortex-antivortex pair unbinding of $\Delta F \approx \pi K \ln(qK/\hbar\xi J)$, where $\xi$ is the vortex radius [34]. The resulting $n_f$ would be proportional to $\exp(-\Delta F/k_B T)$ [32, 33, 35]. Combined, this gives us the low-temperature constitutive relation of [32, 33]

$$\mathbf{j} = \frac{\hbar}{q}\tilde{\sigma}_{\mathrm{dual}}\left(\frac{J}{J_0}\right)^{2T_{\mathrm{BKT}}/T}\mathbf{J} \times \hat{\mathbf{z}} \quad \text{for } T < T_{\mathrm{BKT}}, \tag{6}$$

where $\tilde{\sigma}_{\mathrm{dual}}$ and $J_0$ are phenomenological parameters in units of the dual conductivity and the 2D current density, respectively, below $T_{\mathrm{BKT}}$ with the exponent coming from the famous relation formula $k_B T_{\mathrm{BKT}} = \pi K/2$. Through the DC Josephson relation $\mathbf{E} = \frac{h}{q}\hat{\mathbf{z}} \times \mathbf{j}$, Eqs. (5) and (6) give rise to the experimentally observed [16, 17, 36] change in the DC current-voltage relation at $T = T_{\mathrm{BKT}}$, *i.e.* the exponent in $V \propto I^\alpha$ dropping from $\alpha = 3$ to $\alpha = 1$ [33, 34, 37].

## 3 Dual EM formulation of easy-plane magnets

Both the dual $d = 2 + 1$ EM theory of Eq. (1) and the constitutive relations Eqs. (5) and (6) are applicable to the 2D easy-plane magnetic insulator [30, 38] with the exception for the finite spin lifetime, which we will show to be crucial in spin transport. As their deconfinement drives the magntic BKT transition [2], merons are now the dual particles of Eq. (1), *i.e.* $n$ and $\mathbf{j}$ as the density and the current density, respectively, of merons, with $n \neq 0$ only for $T > T_{\mathrm{BKT}}$. Starting from this identification, we will now explicitly list as Eqs. (2a)-(6a) the spin analogues of the equations Eqs. (2)-(6) for the superconducting films.

First, a meron represents the vortex spin-field configuration - an example being shown in Fig. 1 (a) - and hence carries the quantized spin current vorticity [30, 39–41]. By using the formal analogy (see Appendix A) between the charge current carried by the Cooper pair condensate $\mathbf{J} = qK\boldsymbol{\nabla}\phi/\hbar$ (with $K$ the phase stiffness and $\phi$ the Cooper-pair wavefunction phase) and the spin current carried by the easy-plane order-parameter texture $\mathbf{J}_z^{\mathrm{sp}} = -K\boldsymbol{\nabla}\varphi$ (with $K$ the spin stiffness and $\varphi$ the azimuthal angle of the magnetic order parameter) and by considering that the vorticity is defined in terms of the gradient of the dimensionless phase/angle variables ($\boldsymbol{\nabla}\phi$ and $\boldsymbol{\nabla}\varphi$), we substitute on the right-hand side of Eq. (2) $\hbar\mathbf{J}/qK$ by $\mathbf{J}_z^{\mathrm{sp}}/K$ to obtain

$$n = \frac{1}{2\pi K}\hat{\mathbf{z}} \cdot (\boldsymbol{\nabla} \times \mathbf{J}_z^{\mathrm{sp}}) \tag{2a}$$

($K$ is now the spin stiffness). Second, given that the dual EM wave from Eq. (1) now should be identified with magnons and that the meron vorticity should be conserved due to its topological nature, we now have a straightforward translation of Eq. (3) to

$$\mathbf{j} = \frac{1}{2\pi K}\hat{\mathbf{z}} \times \left( \frac{\partial \mathbf{J}_z^{\mathrm{sp}}}{\partial t} + v^2 \boldsymbol{\nabla} s_z \right), \tag{3a}$$

where the Cooper pair charge density $\rho$ is replaced by the perpendicular spin density $s_z$. Similar analogy also holds for the constitutive relation as the *average* meron mobility is also analogous to the vortex mobility, *i.e.* $\mathbf{v} = w\mu\mathbf{e}$ [42, 43]. While merons in ferromagnet have transverse mobility arising from the core magnetization with a constant Hall angle [30], their average effect cancels out in the absence of an external magnetic field which would give us the zero average core magnetization [44–46]. We hence obtain the third equation for the spin analogue - the Eq. (5) high-temperature constitutive relation in the 2D easy-plane magnet language,

$$\mathbf{j} = \mu n_f \mathbf{J}_z^{\mathrm{sp}} \times \hat{\mathbf{z}} \quad \text{for } T > T_{\mathrm{BKT}}, \tag{5a}$$

where $n_f$ is the combined density of free merons and free antimerons. Likewise, the fourth equation is the Eq. (6) low-temperature constitutive relation,

$$\mathbf{j} = \tilde{\sigma}_{\mathrm{dual}} \left( \frac{J_z^{\mathrm{sp}}}{J_0^{\mathrm{sp}}} \right)^{2T_{\mathrm{BKT}}/T} \mathbf{J}_z^{\mathrm{sp}} \times \hat{\mathbf{z}} \quad \text{for } T < T_{\mathrm{BKT}}, \tag{6a}$$

as $\mathbf{J}_z^{\mathrm{sp}}$ applies a purely Magnus force to each meron on average while the attractive meron-antimeron interaction is logarithmic at long distance. Yet, the exact analogy between the 2D easy-plane magnet and the thin superconducting film stops here, for the spin in the former is not conserved but has a finite lifetime $\tau$ in contrast to the charge in the latter. This modifies the dual Faraday law into [2]

$$-\frac{s_z}{\tau} = \boldsymbol{\nabla} \cdot \mathbf{J}_z^{\mathrm{sp}} + \frac{\partial s_z}{\partial t}. \tag{4a}$$

## 4  Spin transport change at BKT transition

Due to this spin non-conservation, an analysis of the magnetic BKT transport needs to go beyond the local relation between the spin current density and the spin torque gradient [47] and compute the inhomogeneity of the spin current density and / or the spin density. Given that leads are essential features of tranport experiments, Eq. (4a) means that, unlike in the thin superconducting film, the inhomogeneity of both the spin current density and the spin torque gradient is unavoidable. It determines the possibility of the long-distance spin transport.

For the DC spin transport, a qualitative change in the spin current density inhomogeneity occurs at $T = T_{\mathrm{BKT}}$, [3] which limits the spin transport to a finite distance only for $T > T_{\mathrm{BKT}}$ but not for $T < T_{\mathrm{BKT}}$. We first note that when the $T > T_{\mathrm{BKT}}$ finite dual conductivity of Eq. (5a) is inserted into the dual Ampère-Maxwell law of Eq. (3a), the DC terms give us

---

[2]This modification implies the existence of *dissipation* power density $\mathcal{P} \propto s_z'^2$, where $s_z'$ is the non-equilibrium spin density, which has no counterpart in the superfluid / superconductor.

[3]For CrCl$_3$ with its extreme easy-plane anisotropy, $T_{\mathrm{BKT}}$ should be around its estimated magnetic interaction strength $\sim$0.8meV$\approx$9.3K [11].

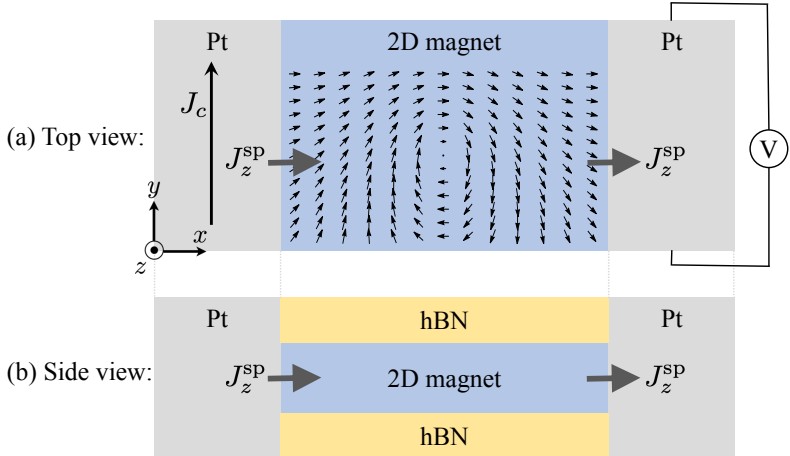

Figure 1: (a) The top and (b) the side view of the proposed experimental setup for spin transport in 2D XY magnets. Spin is transported through 2D XY magnets such as $NiPS_3$ and $CrCl_3$ that are encapsulated by the hexagonal boron nitride (hBN), which is an experimental setup akin to what has already been used for studying 2D Ising magnets, *e.g.* for $CrI_3$ in Refs. [48, 49] and for $CrBr_3$ in Ref. [50]. The injection and the detection of a spin current $J_z^{\text{sp}}$ are performed by using a heavy metal such as Pt as spin-current source and drain (via the spin Hall effect and the inverse spin Hall effect), which is analogous to the experimental realization of the injection and the detection of charge current in monolayer and bilayer graphene using Au *e.g.* Refs. [51, 52].

the spin diffusion, $\mathbf{J}_z^{\text{sp}} = -(v^2/\mu n_f)\boldsymbol{\nabla} s_z$. Diffusive transport, when combined with the finite lifetime as in Eq. (4a), gives rise to the 'mean free path'

$$\lambda_0 = \sqrt{\left(\frac{v^2}{\mu n_f}\right)\tau} = v\sqrt{\frac{\tau}{\mu n_f}}, \tag{7}$$

which in this case means the decay length for the DC spin current [53] from the following equation

$$\mathbf{J}_z^{\text{sp}} = \lambda_0^2 \boldsymbol{\nabla}(\boldsymbol{\nabla} \cdot \mathbf{J}_z^{\text{sp}}) \quad \text{for } T > T_{\text{BKT}}. \tag{8}$$

We can see here that for $T > T_{\text{BKT}}$, the range of spin transport is limited to a length scale that is proportional to the average distance between free merons $\propto n_f^{-1/2}$, which diverges upon approaching $T_{\text{BKT}}$ due to the same singular vanishing of $n_f$ as in the superconducting film [47]. By contrast, below the BKT temperature, we obtain by combining Eq. (6a) with Eqs. (3a), (4a) and retaining only the DC terms,

$$\left(\frac{J_z^{\text{sp}}}{J_0^{\text{sp}}}\right)^{2T_{\text{BKT}}/T} \frac{\mathbf{J}_z^{\text{sp}}}{J_0^{\text{sp}}} = \tilde{\lambda}^2 \boldsymbol{\nabla}\left(\boldsymbol{\nabla} \cdot \frac{\mathbf{J}_z^{\text{sp}}}{J_0^{\text{sp}}}\right) \quad \text{for } T < T_{\text{BKT}}, \tag{9}$$

where $\tilde{\lambda}^2 = v^2\tau/\tilde{\sigma}_{\text{dual}}$. That the power-law ansatz $\mathbf{J}_z^{\text{sp}} = c(x+x_0)^\alpha \hat{\mathbf{x}}$ gives us a solution to this equation with $\alpha = -T/T_{\text{BKT}}$ indicates that the spin current for $T < T_{\text{BKT}}$ decays algebraically rather than exponentially with the distance, giving us the superfluid spin transport. This represents one of the main results of our work: 2D easy-plane magnets support the superfluid spin transport not only at zero temperature [19] but also at finite temperatures despite the

lack of the long-range order so long as free merons are absent. The power-law asymptotic solutions of Eq. (9) that accounts for a realistic spin-current boundary conditions will be discussed below with a concrete experimental setup.

For an experimental setup to detect the predicted behavior of spin transport at the BKT transition, we propose to utilize two heavy-metal leads with strong spin-orbit coupling such as Pt or W (separated by distance $L$) to inject and detect a spin current as shown in Fig. 1; we note that this setup has already been fabricated for the transport measurement of a monolayer vdW material [50]. In this setup, the uniform DC charge current density $J_c$ along the interface (parallel to $\hat{\mathbf{y}}$) in the left lead exerts the interfacial spin torque via the spin Hall effect [54], which gives rise to the spin current flowing in the $x$ direction: $\mathbf{J}_z^{\mathrm{sp}} = \hat{\mathbf{x}} J_z^{\mathrm{sp}}(x)$ (spin-polarized in the $z$ direction). The injected spin is transported through the easy-plane magnet with finite dissipation rooted in the finite spin lifetime as well as the vortex interference. The output spin current from the 2D magnet flowing into the right lead induces the electromotive force via the inverse spin Hall effect [54], which gives rise to the inverse spin Hall voltage signal in the right lead. [4] For the DC case, we have the following boundary condition, which supplements the bulk equations shown in Eq. (8) (for $T > T_{\mathrm{BKT}}$) or Eq. (9) (for $T < T_{\mathrm{BKT}}$):

$$
\begin{aligned}
J_z^{\mathrm{sp}}(0) &= \vartheta J_c - \frac{\hbar g^{\uparrow\downarrow}}{4\pi} \dot{\phi}(0) = \vartheta J_c + \tilde{g} \left. \frac{dJ_z^{\mathrm{sp}}}{dx} \right|_{x=0}, \\
J_z^{\mathrm{sp}}(L) &= \frac{\hbar g^{\uparrow\downarrow}}{4\pi} \dot{\phi}(L) = -\tilde{g} \left. \frac{dJ_z^{\mathrm{sp}}}{dx} \right|_{x=L},
\end{aligned}
\tag{10}
$$

where $\vartheta$ is the effective spin Hall coefficient, $g^{\uparrow\downarrow}$ is the effective interfacial spin-mixing conductance, $\dot{\phi}$ is the local spin precession rate and $\tilde{g} \equiv (\hbar g^{\uparrow\downarrow}/4\pi)(v^2/K)\tau$ parametrizes the spin pumping at the interface within the spin Hall phenomenology [54]. To connect the spin precession rate to the spin current derivative, we used $\dot{\phi} = (v^2/K)s_z$ together with Eq. (4a).

The transition in the spin transport across $T_{\mathrm{BKT}}$ can be seen in Fig. 2, which shows our numerical calculation of the spin current $J_z^{\mathrm{sp}}$ as a function of the distance $L$ [Eqs. (8) and (9)] with the boundary conditions of Eq. (10). Note that the decaying behavior of $J_z^{\mathrm{sp}}(L)$ does not look strikingly different between the $T = 0$ case and the $T = 0.7\,T_{\mathrm{BKT}}$ case, despite the long-range spin ordering that is present in the former but absent in the latter. This result can be supported analytically, as the exact solution for the outgoing spin current at $T = 0$ comes out to be $J_z^{\mathrm{sp}}(L) = \vartheta J_c \tilde{g}/(L + 2\tilde{g})$ [19], while the general asymptotic behavior below the BKT temperature for the outgoing spin current is

$$
J_z^{\mathrm{sp}}(L) \sim \frac{\tilde{g}}{L} L^{-T/T_{\mathrm{BKT}}} \quad \text{for } T < T_{\mathrm{BKT}},
\tag{11}
$$

which we shall derive in Appendix. However, once the temperature is above $T_{\mathrm{BKT}}$, Eq. (8) gives us a qualitatively different asymptotic behavior, an exponential decay

$$
J_z^{\mathrm{sp}}(L) \sim \exp(-L/\lambda_0) \quad \text{for } T > T_{\mathrm{BKT}}.
\tag{12}
$$

This should be readily detectable by the inverse spin Hall voltage in the right lead of Fig. 1, which is proportional to $J_z^{\mathrm{sp}}(L)$. The divergence of $\lambda_0$ just above $T_{\mathrm{BKT}}$, as shown in Eq. (7), should allow us to distinguish the meron contribution obtain here from the thermal magnon contribution.

---

[4]Note that the relation between the interface spin torque and the spin current in heavy-metal / textbar magnet junction is analogous to the relation between the interface voltage and the current in metal / textbar superconductor junction [55].

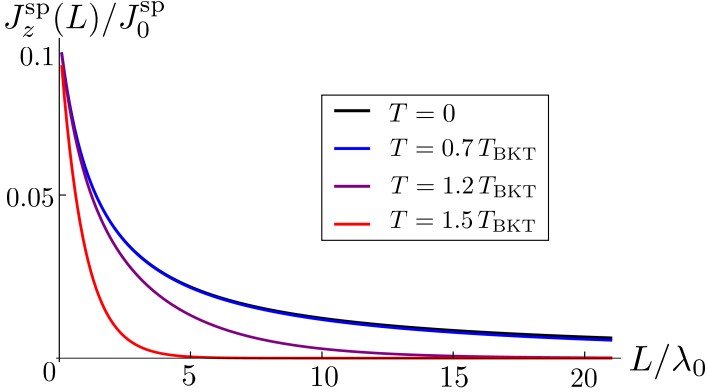

Figure 2: The numerical solution of the differential equations for the bulk spin current spatial variation Eqs. (8) and (9) and the boundary conditions of Eq. (10) for $J_z^{\rm sp}(L)$ and the sample length $L$; the blue, the black, the red, and the yellow curves are for the temperatures $T = 0$, $T = 0.7\,T_{\rm BKT}$, $T = 1.2\,T_{\rm BKT}$, and $T = 1.5\,T_{\rm BKT}$ respectively; following Ref. [47], we have set $\tilde{\lambda} = 0.1\lambda_0 \exp\left[\frac{\pi}{2}(T/T_{\rm BKT} - 1)^{-1/2}\right]$ for $T = 1.2\,T_{\rm BKT}$, and $T = 1.5\,T_{\rm BKT}$.

## 5   Conclusion

Transport signature of a BKT transition should arise from the presence (for $T > T_{\rm BKT}$) or the absence (for $T < T_{\rm BKT}$) of the finite density $n_f$ of topological defects in any 2D XY systems, yet we have shown that its manifestation would be different in 2D easy-plane magnets due to the spin non-conservation of Eq. (4a), which contrasts with thin superconductor / superfluid films possessing the charge / mass conservation of Eq. (4). For the 2D easy-plane magnet, the main impact at $T_{\rm BKT}$ lies in the transport range rather than the disspation, which is present even at the low temperature and is the cause of the spin non-conservation.

We expect our results to be relevant in any systems where the BKT transition occurs but the analogue of the charge conservation does not hold. Recently, the spin superfluidity in the spin-triplet superconductor has been analyzed with the effect of the spin lifetime included [56]. Given the recent advance in fabricating the thin film samples [57–59], this may provide us with yet another venue for detecting the spin transport described in this work.

Lastly, it would be worthwhile to derive a more general dual theory for spin transport which can include the breaking of the U(1) in-plane spin rotational symmetry that has been assumed in this work. Physically, such symmetry breaking may arise from the additional anisotropy within the easy plane, which has been shown to give rise to the critical barrier for superfluid spin transport [18, 19], or from the random anisotropy [60]. Such approach may benefit from taking an alternative perspective within topological hydrodynamics, relying on the conservation of topological charges rather than spins [61].

## Acknowledgements

**Funding information**   We would like to thank Je-Geun Park, Oleg Tchernyshyov, and Yaroslav Tserkovnyak for discussing the preliminary draft of the manuscript with us and Steve Kivelson and Tae Won Noh for motivating this work. We also would like to thank

Young Jun Chang, Jeil Jung, Cheol-Hwan Park, S. Raghu, Jun Ho Son, and Michael Stone for sharing their insights. The work was supported by Brain Pool Plus Program through the National Research Foundation of Korea funded by the Ministry of Science and ICT (Grant No. NRF-2020H1D3A2A03099291) (S.K.K.), the National Research Foundation of Korea funded by the Korea Government via the SRC Center for Quantum Coherence in Condensed Matter (Grant No. NRF-2016R1A5A1008184) (S.K.K.), and the Basic Science Research Program through the National Research Foundation of Korea(NRF) funded by the Ministry of Education(2020R1A2C1007554 and 2018R1A6A1A06024977) (S.B.C.).

# A  Analogy between 2D superconductors and 2D easy-plane magnets

In this section, we discuss the analogous structure between the low-energy dynamics of 2D superconductors and those of 2D easy-plane magnets by closely following the discussion in Ref. [23]. The effective Hamiltonian describing a smooth change of the Cooper pair wavefunction $\psi = \sqrt{\rho}\exp(i\phi)$ is given by

$$H_c = \int dxdy \left[ \frac{K_c(\boldsymbol{\nabla}\phi)^2}{2} + \frac{(\rho - \rho_{\text{eq}})^2}{2C} \right], \tag{13}$$

where $\rho_{\text{eq}}$ is the equilibrium Cooper pair density, $K_c$ is the phase stiffness and $C$ is the capacitance. Here, we truncated the expansion at the leading, quadratic order in the deviations from the equilibrium. The phase and the density are a pair of canonically conjugate variables, and their Hamilton equations are given by

$$\hbar\frac{d\phi}{dt} = -\frac{\delta H}{\delta\rho} = -\frac{\rho - \rho_{\text{eq}}}{C}, \tag{14}$$

$$\frac{d\rho}{dt} = \frac{\delta H}{\hbar\delta\phi} = -\frac{K_c\boldsymbol{\nabla}^2\phi}{\hbar}. \tag{15}$$

The first equation is the Josephson relation with the identification of $(\rho - \rho_{\text{eq}})/C$ as the local non-equilibrium voltage. The second equation is the particle-number continuity equation, from which the expression for the number current can be identified: $K_c\boldsymbol{\nabla}\phi/\hbar$. The corresponding charge current is given by $\mathbf{J}_c = qK_c\boldsymbol{\nabla}\phi/\hbar$, where $q = 2e$ is the charge of a single Cooper pair.

Now, let us turn to the easy-plane magnets. The effective Hamiltonian for the low-energy dynamics of the 2D easy-plane magnet is given by

$$H = \int dxdy \left[ \frac{K_s(\boldsymbol{\nabla}\varphi)^2}{2} + \frac{s_z^2}{2\chi} \right], \tag{16}$$

where $\varphi$ is the azimuthal angle of the order parameter within the $xy$ plane, $s_z$ is the z-component of the spin density, $K_s$ is the spin stiffness, and $\chi$ parametrizes the magnetic susceptibility. In quantum mechanics, the spin density $s_z$ is the generator of the spin rotations within the $xy$ plane, which, in the Hamiltonian formalism, corresponds to that the angle $\varphi$ and the spin density $s_z$ are a pair of canonically conjugate variables. Their Hamilton equations

are given by

$$\frac{d\varphi}{dt} = \frac{\delta H}{\delta s_z} = \frac{s_z}{\chi}, \tag{17}$$

$$\frac{ds_z}{dt} = -\frac{\delta H}{\delta \varphi} = K\boldsymbol{\nabla}^2\varphi, \tag{18}$$

where the spin dissipation is neglected. The first equation describes the spin precession induced by the non-equilibrium spin density, resembling the Josephson relation. The second equation is the spin continuity equation, from which the expression of the spin current is obtained: $\mathbf{J}_z^{\mathrm{sp}} = -K\boldsymbol{\nabla}\varphi$. Note that analogous structure between Eqs. (14, 15) for 2D superconductors and Eqs. (17, 18) for 2D easy-plane magnets. In real magnets, there is always finite spin dissipation and, at the simplest level, it can be accounted for by adding $-s_z/\tau$ to the right-hand side of the second equation, where $\tau$ is the spin lifetime.

By using this analogy between 2D superconductors and 2D easy-plane magnets, a theoretical study of spin transport with the account of the finite spin life time has been undertaken in Refs. [19, 20, 55], but with no consideration of thermal vortices. A non-local spin transport signal over a distance $L$ between a spin-current source and a spin-current detector is shown to decay algebraically $|\mathbf{J}_z^{\mathrm{sp}}| \propto 1/L$ at sufficiently low temperatures much below the BKT transition [19]. In this work, we study spin transport at finite temperatures with full account of thermal vortices, by extending the previous works.

# B   Analytic approximation of the spin current spatial variation

For $T > T_{\mathrm{BKT}}$, Eq. (7) with the boundary condition Eq. (9) of the main text can be solved analytically as

$$J_z^{\mathrm{sp}}(L) = \frac{4\tilde{g}}{\lambda_0}\vartheta J_c \left[(1+\tilde{g}/\lambda_0)^2 e^{L/\lambda_0} + (1-\tilde{g}/\lambda_0)^2 e^{-L/\lambda_0}\right]^{-1}, \tag{19}$$

clearly giving us an exponential decay with $L$ for $L \gg \lambda_0$.

Meanwhile, for $T < T_{\mathrm{BKT}}$, we may use $\tilde{g}/L$ as a small parameter and consider the first-order expansion $J_z^{\mathrm{sp}} = \bar{J}_z^{\mathrm{sp}} + (\tilde{g}/L)\delta J_z^{\mathrm{sp}}$, where $\bar{J}_z^{\mathrm{sp}}$ is the solution of Eq. (8) with the boundary condition Eq. (9) of the main text modified by $\tilde{g} = 0$, i.e. $\bar{J}_z^{\mathrm{sp}}(0) = \vartheta J_c$ and $\bar{J}_z^{\mathrm{sp}}(L) = 0$. This small $\tilde{g}$ limit then would give us

$$J_z^{\mathrm{sp}}(L) = -\tilde{g}\left.\frac{d\bar{J}_z^{\mathrm{sp}}}{dx}\right|_{x=L}. \tag{20}$$

To obtain $\frac{d}{dx}\bar{J}_z^{\mathrm{sp}}(L)$, we note that assuming $\vartheta J_c > 0$, we can take $d^2\bar{J}_z^{\mathrm{sp}}/dx^2 > 0$ and $d\bar{J}_z^{\mathrm{sp}}/dx < 0$ for $0 \leq x \leq L$, and so, by integrating Eq. (8) of the main text, we obtain

$$\tilde{\lambda}\frac{d}{dx}\frac{\bar{J}_z^{\mathrm{sp}}}{J_0^{\mathrm{sp}}} = -\sqrt{\frac{1}{1+T_{\mathrm{BKT}}/T}\left(\frac{\bar{J}_z^{\mathrm{sp}}}{J_0^{\mathrm{sp}}}\right)^{2+2T_{\mathrm{BKT}}/T} + \left|\frac{\tilde{\lambda}\frac{d}{dx}\bar{J}_z^{\mathrm{sp}}(L)}{J_0^{\mathrm{sp}}}\right|^2}.$$

We use $\int_0^\infty dx/\sqrt{x^\alpha + 1} = \Gamma(1/2 - 1/\alpha)\Gamma(1+1/\alpha)/\sqrt{\pi}$ for $\alpha > 2$ to derive

$$\lim_{L/\tilde{\lambda}\to\infty}\frac{L}{\tilde{\lambda}}\left|\frac{\tilde{\lambda}\frac{d}{dx}\bar{J}_z^{\mathrm{sp}}(L)}{J_0^{\mathrm{sp}}}\right|^{\frac{1}{1+T/T_{\mathrm{BKT}}}} = \frac{1}{\sqrt{\pi}}\left(1+\frac{T_{\mathrm{BKT}}}{T}\right)^{\frac{1}{2+2T_{\mathrm{BKT}}/T}}\Gamma\left(\frac{1}{2+2T/T_{\mathrm{BKT}}}\right)\Gamma\left(\frac{2+3T/T_{\mathrm{BKT}}}{2+2T/T_{\mathrm{BKT}}}\right). \tag{21}$$

Eqs. (20) and (21) together gives us Eq. (11) of the main text.

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
