# Peer review of "Transport signature of the magnetic Berezinskii-Kosterlitz-Thouless transition"

_SciPost Physics_

## Round 2 · Referee Report · Anonymous (Referee 1) · 2020-11-12

Strengths

  1. The magnetic BKT transition is a critical phenomena that needs precise formulation in the modern context of spin transport and spin superfluidity in 2+1 space -time. In this the paper provides a comprehensive analysis.

  2. The paper is well constructed, I appreciate the inroduction of the effective field theory in the context of the well studied superconducting vortex BKT.

  3. The experimental setup seems feasible. There are probably some challenges lying in how to block out thermal noise and diffusion when the system is heated. As such the analysis is devoid of phonons which makes it hard to undestand those diffusive processes. However, given that the transition is topological the signatures will be present.

Weaknesses

  1. I am not sure what the authors mean by 'in-plane isotropy' on page 1.

  2. There are some issues with the equation numbering in section 3.

  3. There are minor grammatical errors. For instance a rather liberal use of 'the'.

Report

There are no obvious errors, but I would like some clarifications especially with regard to the duality map (lagrangian used).

  1. Regarding Eq. 4 it is probably more apt to call this the Bianchi identity. Identifying this conserved U(1) current is the first step to contructing the dual Lagrangian. Identifyng this as the Faraday law is like putting the horse before the cart.

  2. The same equation has to bear the brunt of the non-conservation of spin current in Eq. 4a. Regarding this equation, it seems correct to incorporate the spin flip process as some sort of damping parameter. What worries me is that this kills the Bianchi identity which is needed to generate the dual EM lagrangian of Eq.1. I believe, although it is not obvious, that the situation is probably remedied by including a Rayleigh term of the form $\alpha(\partial_t\varphi)^2$.

  3. The authors worry about a small break in the U(1) symmetry it turns out that this really does not matter in the low energy theory (vortex cores not overlapping) as long as the easy plane anisotropy dominates over the other anisotropies, see the discussion in Phys. Rev. B 102, 144417. Here the low energy theory is still U(1) with the six fold anisotropy adding a small perturbation.

Overall I think this is a solid piece of work and I reccomend publication with some terminology revisions.

  • validity: high
  • significance: good
  • originality: good
  • clarity: good
  • formatting: excellent
  • grammar: good

Author:  Suk Bum Chung  on 2021-02-22  [id 1258]

(in reply to Report 1 on 2020-11-12)
Category:
answer to question
correction

We thank Referee for the appreciative review.

Given that the Faraday law itself is not really derived from the Maxwell Langrangian, Referee was right to ask for there should be a prior mention of the Bianchi identity for Eq 4; we have revised our text accordingly. As he/she pointed out, the spin non-conservation of Eq 4a requires a Rayleigh dissipation term, explanation of which has been added to the existing footnote to Eq 4a.
In addition, we added a footnote to the penultimate sentence of the Conclusion section to note that the in-plane hexagonal anisotropy does not qualitatively change the BKT transition.

---

## Round 2 · Referee Report · Anonymous (Referee 2) · 2021-2-17

Strengths

1) I acknowledge that the authors explain first the validity of their theoretical approach (Dual EM theory) using superconducting films and thus a problem based only on charge current. The comparison between superconducting vortices and merons become thus very clear for the reader (spin is not conserved in the latter case).
2) Their main message is explained very clear throughout the manuscript: Spin transport decay algebraically with distance below BKT, and exponentially above. For T>TBKT, spin transport coherence is proportional to average distance between merons. For T<TBKT, the superfluid spin transport regime at finite temperatures exists as long as free merons are absent. The fingerprint of the BKT transition in the spin transport characteristics is hence well defined.
3) The authors give a plausible experimental approach where their predicted effect can be measure in a device geometry using spin injection and detection via spin hall effect. This will make a great impact for experimentalists working with low-dimensional magnets.

Report

The manuscript by Kim.et.al. deals with the theoretical investigation of the spin transport characteristics of easy-plane magnetic systems. The main finding is to define a fingerprint of the BKT transition in the spin transport behavior as a function of distance (power-law dependence below and exponentially above the BKT transition temperature), using a theoretical approach based on the dual EM theory. In addition, they suggest an experimental setup on how to measure the predicted phenomenology, in terms of device geometry and material selection. The paper is well written and organized and merits publication in SciPost after minor revisions.

Requested changes

1) For the sake of completeness, the authors should acknowledge and cite in their introduction the existence of a BKT transition in adsorbed atoms and reconstructions on surfaces (e.g. Baek, et.al. Critical behavior of the p(2×1)-O/W(110) system. Phys. Rev. B 47, 8461(1993), as well as recent experimental advances in the fabrication of monolayer CrCl3 with ferromagnetic 2DXY behavior https://arxiv.org/abs/2006.07605 (more relevant than the cited previous reports that relied on the layered antiferromagnetic bulk phase of CrCl3). Especially the latter is important to motivate that there are already existing platforms for the theoretical predictions to be validated.

2) In Figure 2, it would be instructive to compute/map more temperatures close to TBKT (e.g. 0.9, 0.95 TBKT), to see where a substantial difference (departure) from the zero-temperature spin transport behaviour can be seen. In addition, since the 2DXY behavior is experimentally attained at temperatures very close to the phase transition, this information would be more relevant (for a Review, see Vaz. et.al, Reports Prog. Phys. 71, 056501 (2008) and for specific critical exponent determination in a CrCl3 monolayer see https://arxiv.org/abs/2006.07605)

3) To support a more conclusive picture, could the authors comment on the evolution of the free meron density as a function of TBKT, or to define a practical temperature window in terms of TBKT where free merons have a substantial contribution to the distance-dependent spin current transport? This would be a valuable information on top of what is given in Figure 2.

  • validity: high
  • significance: top
  • originality: high
  • clarity: high
  • formatting: excellent
  • grammar: good

Author:  Suk Bum Chung  on 2021-02-22  [id 1257]

(in reply to Report 2 on 2021-02-17)
Category:
answer to question
correction

We are most grateful to Referee for the appreciative report.
Concerning Fig 2, we agree with Referee that it is important to have plots for temperature closer to the BKT temperature. In the revised version, we have revised our figure and caption accordingly and noted in our caption that the temperature window where the free meron density have shown very close agreement with the BKT critical dependence on temperature in a recent numerical calculation (Phys Rev B 100 144416 listed as Ref [51]).
We also would like to thank Referee for the references that he/she introduced to us, in particular the recent CrCl3 experiment (https://arxiv.org/abs/2006.07605).

---

## Editorial Decision

resubmitted